# An Assessment of the Ecological Landscape Quality (ELQ) of Nature-Based Solutions (NBS) Based on Existing Elements of Green and Blue Infrastructure (GBI)

**Barbara Sowińska-Świerkosz [1]**, **Julia Wójcik-Madej [1]** and **Malwina Michalik-Śnieżek [2],\***

[1] Department of Hydrobiology and Ecosystems Protections, University of Life Sciences in Lublin, 20-262 Lublin, Poland; barbara.sowinska@wp.pl (B.S.-Ś.); julia.wojcik3@wp.pl (J.W.-M.)

[2] Department of Grassland and Landscape Shaping, University of Life Sciences in Lublin, 20-950 Lublin, Poland

\* Correspondence: malwina.sniezek@up.lublin.pl; Tel.: +48-503-508-961

**Abstract:** Nature-based solutions (NBS) positively impact ecological landscape quality (ELQ) by providing multiple benefits, including enhancing natural capital, promoting biodiversity, mitigating water runoff, increasing water retention, and contributing to climate change adaptations and carbon sequestration. To analyze the specific contribution of different NBS types, this study assessed 14 ELQ indicators based on the application of spatial data. Five NBS based on existing elements of green and blue infrastructure (GBI) were analyzed at the city level (Lublin, Poland), including parks (UPs), forests (UFs), water bodies (UWs), allotment gardens (AGs), and woods (Ws). The analysis revealed that different NBS contribute in contrasting ways to the improvement of various dimensions of ELQ. UFs made the biggest contribution to the maintenance of ecological processes and stability, as well as to aesthetic values. Ws together with AGs were crucial to maintaining a high level of diversity at the landscape scale and also contributed to preserving the ecological structure. UWs and UPs had no outstanding impact on ELQ, mainly due to their high level of anthropogenic transformation. The application of spatial indicators proved useful in providing approximate information on the ecological values of different types of NBS when other data types were either unavailable or were only available at a high cost and with considerable time and effort.

**Keywords:** urban ecology; ecological indicators; green/blue infrastructure; landscape quality; nature based solutions; remote-sensing





## 1. Introduction

Nature-based solutions (NBS) are a multidisciplinary umbrella concept that links social and economic benefits with the notion of 'nature' [1,2]. These solutions should be either inspired by, supported by, or copied from nature; furthermore, the use of nature should be treated as a priority and not a supplement to conventional infrastructure [3,4]. NBS should also be cost-effective, resource-efficient, and locally adopted [5] and lead to multiple benefits by supporting sustainable development [6]. Moreover, stakeholders' participation, policies, and management capability and performance in the long term are emphasized as necessary factors for considering a given green solution as NBS [1,3–5]. Consequently, NBS can be defined as solutions that are oriented to urgent problem(s) that simultaneously address environmental, social, and economic challenges by the use of plants, water, and/or chemical processes, are inspired by nature, provide multifunctional benefits, and have considerable management potential and economic efficiency [7]. The broad goals of NBS reflect the idea of sustainability and resilience by searching for innovative solutions to manage the natural environment in a way that balances benefits for both nature and society [8]. By working with nature rather than against it, communities can develop and implement solutions that will pave the way for sustainable city development [2]. The concept of NBS is congruent

with many 'Sustainable Development Goals (SDGs)' [9] because, by working with nature, effective solutions for peaceful, prosperous, and equitable societies can be developed. Nature provides people with resources, including the food, air, water, and energy required for peoples' health and well-being. Additionally, nature can be harnessed to introduce solutions to the challenges set out in the SDGs, benefiting environmental, social, and economic outcomes [10]. The handbook prepared by Dumitru and Wendling [4] in 2021 described the 12 categories of societal challenges that can be addressed by NBS, thus linking the scope of green interventions with SDGs. As a result, NBS were addressed on the base of a triad of societal challenges for people (e.g., place regeneration, knowledge and social capacity buildings, participatory planning and governance, social justice and cohesion, wealth and wellbeing), the planet (e.g., climate resilience, water management, green space management, biodiversity, air quality), and prosperity (natural and climate hazards, new economic opportunities, green jobs), which are the pillars of sustainable development. NBS can contribute to achieving SDGs and delivering sustainable development for everyone [10]. This statement was also supported by many previous studies, including SDGs, such as tackling poverty (e.g., urban gardens), good health and well-being (e.g., urban forests), reducing inequalities (e.g., open green spaces), clean water (e.g., constructed wetlands), climate actions (e.g., avenue of trees), life on land (e.g., plant shelter belts), and industry, innovation, and infrastructure (e.g., green roofs) [11–13].

The NBS concept is closely connected with many existing green solutions, including green and blue infrastructure (GBI), which, according to the European Commission [1], is defined as "a strategically planned network of natural and semi-natural areas with other environmental features designed and managed to deliver a wide range of ecosystem service ( . . . )". GBI includes all natural and semi-natural elements that form a green-blue network and refer to landscape elements on various spatial scale levels. GBI includes small-scale elements, such as hedgerows, bushes, and ponds, and large-scale elements, such as urban gardens, forests, and parks. Therefore, according to a report by the EC [1], actions that enhance the existing green (historical) infrastructure to resolve urgent challenges should be treated as a sort of NBS, alongside more novel solutions, such as constructed wetlands and green roofs and walls. Elements of urban GBI may be created as independent NBS actions, including interventions, such as ensuring continuity of the ecological network by the application of green belts, implementation of vegetated filter strips, or the creation of surface wetlands. In the case of the low efficiency of existing elements of GBI, they may be enlarged or fitted with new green and grey elements, such as semi-permeable surfaces, water collecting tanks, infiltration planters, tree boxes, or apiaries [4,12].

Elements of GBI, however, constitute one of the types of NBS interventions. According to the typology proposed by Eggermont et al. [14] and further developed by Somarakis et al. [15], NBS are divided into three general types depending on the level of human intervention. Type 1 includes minimal intervention in ecosystems, meaning better use of protected/natural ecosystems, which includes protection and conservation strategies, such as the maintenance or enhancement of natural wetlands, the control of urban expansion, and regular monitoring of physical, chemical, or biological indicators [4]. Type 2 deals with extensive or intensive management approaches that develop sustainable and multi-functional ecosystems and landscapes, which improves the delivery of multiple benefits. This type includes actions of urban green space management, such as the creation and preservation of habitats and shelters to support biodiversity, sustainable fertilizer use or composting of organic wastes, and reuse of composted materials [4]. Type 3 features the highest level of human intervention and includes the design and management of new ecosystems, such as green spaces, permaculture, green roofs and walls, surface wetlands, and infiltration basins [4].

However, different types of NBS affect ecological landscape quality (ELQ) differently. ELQ is understood as an ecological condition of a given landscape, being an effect of superimposing upon a set of environmental components, processes, and phenomena that are subjected to a direct outcome or a side effect of human activity [16]. ELQ is related

to physical environmental characteristics, such as soil, water, air, and plants, and can encompass many elements, including environmental pollution and cleanliness, structural and functional connectivity between habitats, and biodiversity at the plant community level [17]. Considering the landscape scale, ecological quality depends on the type, variety, and density of the natural and anthropogenic elements existing within a specific context and on the level of quality associated with these elements [18]. ELQ can be assessed both: (1) from the 'ground' perspective, by the direct, in-situ analysis of quality of different components of the environment, such as water, air, and soil (e.g., soil drilling, water samples, weather stations); and (2) from 'a bird's-eye view' perspective, by analyzing the structure and connectivity of land use/land cover (LU/LC) patches with digital assessment through GIS analysis based on remote-sensed data acquisition techniques [19]. Given that most NBS projects deal with conservation, restoration, and cultivation goals [12], they have a multitude of positive effects on all environmental components and generally contribute to improving ELQ and promoting 'good-quality' landscapes by utilizing more natural features and processes in landscapes and seascapes [1]. These include enhancing natural capital, promoting biodiversity, mitigating water runoff, increasing water retention and infiltration, contributing to climate change adaptations and carbon sequestration, reducing emissions, mitigating the urban heat island effect, and removing pollutants [20–22]. NBS may also improve the connectivity of biologically active areas at the city level and provide a bridge between urban/peri-urban areas and natural areas. NBS are particularly applied when considering 'landscape-scale' initiatives, such as regional/national strategy for afforestation or flood protection. In addition, the area and configuration of vegetation patches influence the stability of ecological processes and affect the amount of carbon capture [3].

The main aim of this research was to assess the ELQ of diverse types of NBS on a city scale (Lublin, Poland). The assessment was executed in relation to NBS actions on the design and management of semi-natural ecosystems, including urban parks, urban forests, urban water, allotment gardens, and wooded areas, representing existing elements of GBI. The practical aim was to indicate which of the existing NBS actions on the Lublin scale makes the greatest contribution to improving the ecological state of the urban landscape and thus sustainable urban development. Therefore, a set of landscape-based indicators, representing the four main aspects that are crucial for assessing ecological quality at the landscape scale, were calculated: (1) maintenance of ecological processes and ecosystem stability; (2) diversity; (3) continuity of ecological structures; and (4) aesthetic landscape values.

## 2. Materials and Methods

The study consisted of three main stages: (1) identifying types of NBS' functions in the analyzed area; (2) spatially detecting the identified NBS within the study area; and (3) calculating the ELQ indicators (Figure 1).

### 2.1. Study Area

The city of Lublin is a cultural, scientific, and economic center and communication hub in eastern Poland. It is the capital of the Lubelskie Voivodeship. In 2020, the population of Lublin reached almost 340,000 citizens. Lublin is characterized by a varied topography, a low river network density, and a moderately developed natural structure dominated by forests, which constitute 11% of the city's area. The ecological system of the city is inextricably linked with three river valleys, which also represent the main ecological corridors. The most important of these corridors is the Bystrzyca River valley, on which a retention reservoir was built in the 1970s (Zalew Zemborzycki) [23]. It is located in the southern part of the contemporary city. In addition to river valleys, there are dry valleys in the city that have been developed as urban parks or wooded areas. Allotment gardens are another important element of Lublin's green infrastructure, patches of which are located throughout the city (Figure 2).

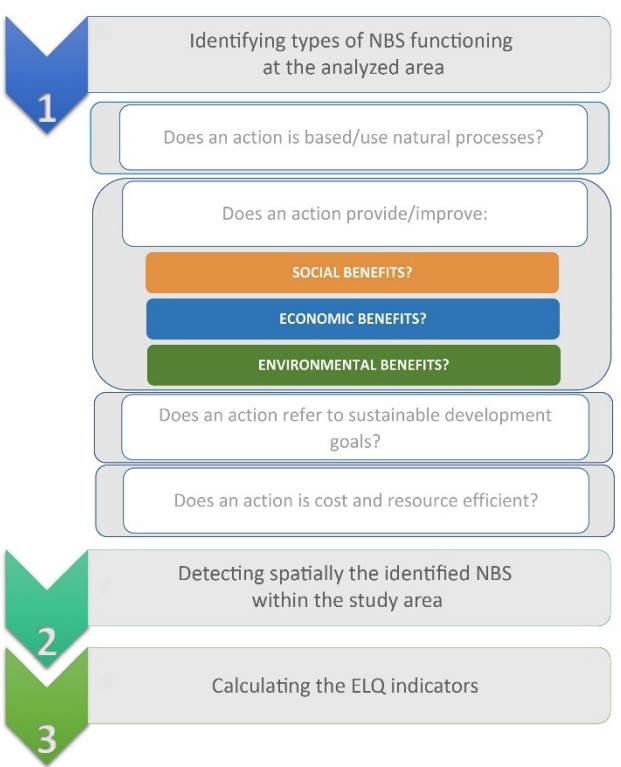

**Figure 1.** Methodological framework.

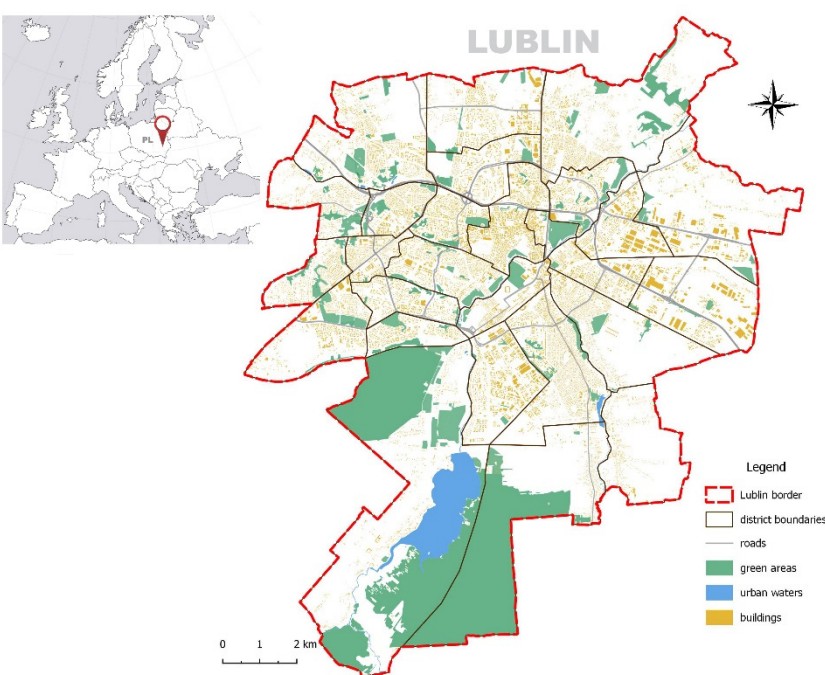

**Figure 2.** Location of the study area and spatial structure of the city.

### 2.2. Identification of NBS on the Lublin City Scale

To identify NBS on the Lublin city scale, the exclusion approach was adopted. Based on the list of NBS types proposed by Dumitru and Wendling [4], the following were determined: (1) whether a given general or particular type of NBS operates in the analyzed area; (2) the presence of spatial characteristics that enable the calculation of landscape-based indicators; and (3) the effectiveness and efficiency of NBS. The procedure consisted of three levels (L) corresponding to (L1) general NBS types [14]; (L2) specific forms has been based

on NBS according to Dumitru and Wendling [4]; and (L3) a particular NBS performance assessment based on the following questions: Is the analyzed solution based on or using natural processes?; Does an analyzed solution provide or improve social, environmental, and/or economic benefits?; Does an analyzed solution refer to the SDGs?; Is the solution cost- and resource-efficient? [7] (Table 1).

**Table 1.** Three criteria levels for the identification of nature-based solutions (NBS) on the city scale (Lublin, Poland).

| |
|---|
| **Level 1:** General three types of NBS based on the typology proposed by Eggermont et al. [14] and further developed by Somarakis et al. [15] |
| **Type 1:** Minimal intervention in ecosystems indicating better use of protected/natural ecosystems (protection and conservation strategies). |
| **Type 2:** Extensive or intensive management approaches that develop sustainable and multifunctional ecosystems and landscapes that improve the delivery of multiple benefits (actions of urban green space management). |
| **Type 3:** Design and management of new ecosystems (creation of artificial or semi-natural ecosystems). |
| **Exclusion criterion**: In the aim to calculate ELQ indicators in the study, all solutions classified as type 1 and 2 were rejected as they were based on different management and monitoring approaches (e.g., controlling urban expansion, regular monitoring of physical, chemical or biological indicators, integrated pest/weed management, sustainable fertilizer use), rather than on solutions that have a material form reflected by a given structure of land cover form that can be detected and measured based on the remote sensing and GIS approach. |
| **Level 2:** Specific forms of NBS according to the list proposed by Dumitru and Wending [4]. |
| **Type 3 NBS:** (1) Green space—multifunctional open space characterized by natural vegetation and permeable surfaces; (2) Trees and shrubs; (3) Soil conservation and quality management; (4) Blue-green space establishment or restoration; (5) Green built environment; (6) Natural or semi-natural water storage and transport structures; (7) Infiltration, filtration, and biofiltration structures.) |
| **Exclusion criterion:** Solutions were screened to detect interventions that: (1) are composed of features that can be detected by remote sensing and are composed of greenery (in the aim to calculate ELQ indicators, which are based on the land cover structure) and (2) are present in the study area. |
| **Level 3:** Specific form of NBS performance based on questions proposed by Sowińska-Świerkosz and García [7]. |
| **Forms of type 3 NBS fulfilling the Level 2 criterion:** Urban parks; Urban gardens; Green strips; Green transport track; Forests; Street trees; Urban water bodies |
| **Exclusion criterion:** The performance questions measuring to what extend a particular solution type existed in the Lublin city structure fulfilled the formal requirements to recognize it as a NBS were adopted. These questions refer to the four core ideas for clarifying the NBS concept proposed by Sowińska-Świerkosz and García [7]. |
| **Is the analyzed solution based on or using natural processes?** |
| NBS refers to actions that 'use nature' or are 'powered by nature' [1]. The conscious use of plants and/or water must be a priority and not a supplement to conventional infrastructure [23]. As there is no consensus on the scope of human intervention on NBS [14,24,25], minimal or no-intervention management approaches that involve some intervention, as well as intrusive ecosystem management approaches, may be treated as NBS. |
| **Does an analyzed solution provide or improve social, environmental, and/or economic benefits?** |
| The second criterion is based on three questions proposed by the EC [2] to define whether an action can or cannot be framed as an NBS. These questions pertain to different types of benefits: social, environmental, and economic. The fourth question—'Does action have a net benefit for biodiversity?'—was not included in the procedure to avoid restricting types of action to those that focus on biodiversity. |
| **Does an analyzed solution refer to the Sustainable Development Goals?** |
| The third criterion reflects the challenge orientation of NBS, seeking to alleviate a well-defined environmental, social, and economic challenge. This challenge-orientation means that it aims to provide a solution to a previously detected problem [26]. This aspect is crucial to distinguishing among NBS and other green interventions [18]. Considering the present study's aim—whether a given action relates to the idea of sustainable urbanization—was determined by indicating the Sustainable Development Goal [27] to which it contributes. |
| **Is solution cost- and resource-efficient?** |
| The fourth criterion relates to the efficiency of NBS actions in terms of costs and resources [5,8,28], that is, the cost of a solution's implementation, management, monitoring and damage over a certain timeframe should not exceed the potential environmental and social benefits. In addition, economic viability refers to the promotion of renewable sources of energy, the use of rainwater or treated water instead of drinking water to irrigate and maintain solutions, and the re-use of materials [3]. |

### 2.3. Spatial Detection of Identified NBS within the Study Area

The identified solutions were spatially detected based on the Polish Database of Topographic Objects (BDOT) (vector format), and the land cover classification [29] was provided and updated to Orthophotomap (pixel size 0.5 m, 2019). To ensure the provision of multiple benefits, both to the environment and citizens, as the core idea of the NBS concept, size and accessibility criterion were adopted for each of the specific NBS types identified at the first stage of the research.

### 2.4. Calculation of ELQ Indicators

Different landscape metrics representing the spatial structure of the landscape were selected to measure the four main aspects crucial for assessing ecological quality at the landscape scale: (1) maintenance of ecological processes and ecosystem stability; (2) diversity; (3) continuity of ecological structures; and (4) aesthetic landscape values [30] (Table 2).

In relation to the maintenance of ecological processes and ecosystem stability, the patchy structure reflected by a set area (MPA), density (PD, ED), and shape (FRAC) may be treated as an indicator of the function of different ecosystems, as well as ecological stability [31]. Indices, such as biologically active area and percentage of areas occupied by a given LC form, also suggest the level of anthropogenic transformation and thus indirectly determine the level of the natural state of the environment, as one of the main aspects ensuring ecosystem stability [30].

To reflect diversity, landscape metrics were used to sample patches of habitats based on the land cover forms, rather than sampling species [11]. This approach assumes that the physical complexity and spatial organization of LC forms, to some degree, reflect the species diversity of a given area. Moreover, changes in land-use structure, especially fragmentation (COHESION), and the area occupied by anthropogenic LC forms (AT) are considered major contributors to plant diversity decline. Landscape structure also determines the diversity of biophysical conditions [32]. Therefore, topographic factors, such as SLOPE, have been shown to be appropriate prediction factors of plant species diversity [33].

To measure the continuity of ecological structures, different indicators measuring patch isolation and the degree of fragmentation were selected. These measures directly indicate the continuity among different LC patches that constitute the habitats of different species [34]. Habitat fragmentation primarily results in an increase in the number of small-size patches and therefore may be effectively quantified with MPA, PD and ED metrics. Additionally, the density of ecological barriers reflects whether the movement of migratory species is affected by the presence of communication routes.

Objective aesthetic landscape values include the naturalness and harmony of land cover patches [30]. As people generally perceived natural and semi-natural forms as well as greenery as having a positive impact on aesthetic values [35], indices on the level of anthropogenic landscape transformation (BAA, AT, ECOLBAR) and fractal index (FRAC) were selected. The letter reflects the shape of the land cover patches; square or almost square patches are typical for human-made structures and natural types of land cover having more irregular shapes and softer boundaries.

All landscape indicators, except slope, were calculated based on the Polish Database of topographic objects [29] updated to Orthophotomap (pixel size 0.5 m, 2019). To calculate AT indices, data on the state of anthropogenic transformation was ascribed to each of the land cover (LC) patches in the attribute table. The following classification was adopted: (1) natural LC forms: non-transformed water, peat-bogs, and meadows and protected forests; (2) semi-natural LC forms: transformed water, peat-bogs, and meadows and economy forests; (3) anthropogenic LC forms type 1: fields and orchards; (4) anthropogenic LC forms type 2: areas without vegetation, that is, buildings and roads [16]. Data on MPA, PLAND, BAA, AT and ECOLBAR were extracted from the attribute tables of BDOT shape files using QGIS software. Values of PD, ED, LPI, FRAC_MN and COHESION were calculated using FRAGSTATS 4.2. (Spatial Pattern Analysis Program for Categorical Maps) software and the moving window method [36]. The resolution of the raster used as an input to the software was 1 m. The slope was calculated based on the Numerical Terrain Model (NTM; grid interval of at least 100 m). To perform slope analysis, the contour lines were first converted into a raster DEM, and then the slope tool (Raster -> Terrain Analysis -> Slope) was applied. QGIS software was used.

As autocorrelation strongly affects environmental analysis, especially when applied composite indexes, correlation analysis of these indicators was performed to avoid duplication and double counting. This analysis is based on the results of a Spearman's rank correlation ($p = 0.05$) for each pair of indices. If the obtained absolute correlation coefficient amounted to 0.9 or more, only one of the two indices was retained. As a result, 14 indicators were analyzed further (Table 3).

**Table 2.** Calculated ELQ indicators.

| Abbreviation | Indicator Name | Unit | Formula/Description | Reference to Dimensions of ELQ |
|---|---|---|---|---|
| MPA | Mean Patch Area | ha | – | S; C |
| PLAND | Percentage of landscape occupied by a given LC form | % | – | S; D; C |
| BAA | Biologically Active Area | % | As BAA was considering areas covered by vegetation and water | S; C; A |
| $AT_{natural/semi-ntural/antr\ no\ 1/antro\ no\ 2}$ | Percentage of areas occupied by patches of different levels of anthropogenic transformation | % | Levels of anthropogenic transformation of LC forms according to Sowińska-Świerkosz and Michalik-Śnieżek [16] | S; A; D; C |
| ECOLBAR | Density of Ecological Barriers | m/ha | As Ecological Barriers were considering paved roads | S; C; A |
| PD | Patch Density | nos/100 ha | $PD = \frac{n_i}{A}(10,000)(100)$ <br> $n_i$ = number of patches in the landscape of patch type (class) $i$ <br> $A$ = total landscape area (m$^2$) | S; D; C |
| ED | Edge Density | m/ha | $LSI = \frac{.25\sum_{k=1}^{m} e_{ik}*}{\sqrt{A}}$ <br> $e_{ik}$ = total length (m) of edge in landscape between patch types * (classes) $i$ and $k$; includes the entire landscape boundary and some or all background edge segments involving class $i$ <br> $A$ = total landscape area (m$^2$) | S; D; C |
| LPI | Largest Patch Index | % | $LPI = \frac{max_{j=1}^{n}(a_{ij})}{A}(100)$ <br> $a_{ij}$ = area (m) of patch $ij$. <br> $A$ = total landscape area (m$^2$) | S; D |
| FRAC_MN | Mean Fractal Dimension Index | – | $FRAC = \frac{2\ln(.25p_{ij})}{\ln a_{ij}}$ <br> $p_{ij}$ = perimeter (m) of patch $ij$ <br> $a_{ij}$ = area (m$^2$) of patch $ij$ | S; A |
| COHESION | Patch Cohesion Index | – | $COHESION = \left[1 - \frac{\sum_{j=1}^{n} p_{ij}*}{\sum_{j=1}^{n} p_{ij}*\sqrt{a_{ij}*}}\right] * \left[1 - \frac{1}{\sqrt{Z}}\right]^{-1} * (100)$ <br> $p_{ij}$ = perimeter of patch $ij$ in terms of number of cell surfaces. <br> $a_{ij}$ = area of patch $ij$ in terms of number of cells. <br> $Z$ = total number of cells in the landscape | S; D; C |
| SLOPE | Slope | % | – | D |

**Note:** S: Maintenance of ecological processes and ecosystem stability; D: Diversity; C: Continuity of ecological structures; A: Aesthetic landscape values.

Table 3. Results of level 3 excluded criterion application: performance on NBS based on questions proposed by Sowińska-Świerkosz and García [7].

| Does an Action Use Nature/Natural Process? | Does an Action Provide/Improve Social Benefits? | Does an Action Provide/Improve Environmental Benefits? | Does an Action Provide/Improve Economic Benefits? | Does an Action Refer to the Sustainable Development Goals? | Is an Action Cost and Resource-Efficient? |
|---|---|---|---|---|---|
| **Urban parks (UPs) [21,37,38]** | | | | | |
| YES | YES | YES | YES | YES | PARTIALLY |
| UPs consist of lawns, trees, shrubs and flowers; paved paths and playgrounds are gradually replaced by semi-permeable or green surfaces. | UPs possess recreational and historical value; they are of great ornamental and perception value; UPs are meeting places and offer contact with nature. | UPs contribute to carbon sequestration, seed dispersal, erosion prevention, water purification, air purification, habitat quality and noise reduction. | UPs provide benefits in terms of the economic development of the community and tourism. | • Good health and well-being; • sustainable cities; • life on land; climate actions; reduce inequalities. | Maintenance of UPs requires continuous and usually costly actions such as planting, watering and mowing, restoration and repair of elements of small architecture, lighting and cleaning. |
| **Urban forests (UFs) [39–41]** | | | | | |
| YES | YES | YES | YES | YES | YES |
| UFs consist of trees and shrubs; paved surfaces such as parking and roads are a rarity. | UFs offer recreational functions and contact with nature; they contribute to psychological and physical health; they have educational, research and teaching functions. | UFs contribute to air purification, global climate regulation, urban temperature regulation, noise reduction and runoff mitigation. | UFs provide benefits in terms of the economic development of the community and tourism. | • Good health and well-being; • sustainable cities; • life on land; climate actions; • reduce inequalities. | UFs require a minimum level of human intervention and do not involve the use of electricity and running water. |
| **Urban waters (UWs) [42–45]** | | | | | |
| YES | YES | YES | YES | YES | PARTIALLY |
| Both natural and artificial UWs function based on natural processes. | UWs support aquatic life, enable relaxation and improve physical and mental health; they possess intangible values to citizens such as amenities and a sense of place for fishing communities; they are of great aesthetic value. | UWs support biodiversity, freshwater storage, hydrological balance, climate regulation, flood protection, water purification and oxygen production and fertility; they provide a physical refugium from predation and are used as nursery and feeding areas. | UWs are a source of food provisioning, energy production and drinking water supply; they contribute to flood hazard reduction; they are used in industry and agriculture production | • No poverty; • good health and well-being; • sustainable cities; climate actions; life below water; • life on land. | UWs' efficiency depends on the case study. There are resource-efficient water bodies and water bodies, especially dam reservoirs, which require maintenance and large amounts of money. |

<div align="center">

**Table 3.** *Cont.*

</div>

| Does an Action Use Nature/Natural Process? | Does an Action Provide/Improve Social Benefits? | Does an Action Provide/Improve Environmental Benefits? | Does an Action Provide/Improve Economic Benefits? | Does an Action Refer to the Sustainable Development Goals? | Is an Action Cost and Resource-Efficient? |
|---|---|---|---|---|---|
| **Allotment gardens (AGs) [16,46–58]** | | | | | |
| YES | YES | YES | YES | YES | PARTIALLY |
| The key "building material" of AGs is greenery: edible and flourishing plants, fruit trees, herbs and grasses and in some cases also water structures. | AGs provide a source of relaxation, hobbies and contact with nature; they promote outdoor activities. | AGs provide a source of relaxation, hobbies and contact with nature; they promote outdoor activities. | AGs provide low-cost food and have a positive impact on reducing medical expenses. | <ul><li>Tackling poverty; good health and well-being;</li><li>clean water;</li><li>sustainable cities;</li><li>responsible consumption;</li><li>life on land.</li></ul> | Most AGs possess rainwater recovery devices and composters and are not equipped with electricity; some AGs possess photovoltaic panels and re-used building materials. |
| **Woods(Ws) [11,49,50]** | | | | | |
| YES | YES | YES | YES | YES | YES |
| Trees and shrubs are the only materials. | Ws contribute to noise reduction; they increase walkability through providing greater feelings of safety; they are of great ornamental, aesthetic and amenity value; they are of great recreational value when other forms of greenery are not available. | Ws reduce pollutant emissions, improve surface run-off, improve soundscape quality, create habitats and have a shading and cooling effect. | Wide strips of trees along roads can to some extent replace expensive acoustic screens; they may produce fruits. | <ul><li>Good health and well-being;</li><li>sustainable cities;</li><li>climate actions;</li><li>life on land.</li></ul> | Ws require watering only during the first few years after planting; after these periods they are self-sufficient. |

## 3. Results

### 3.1. Identification of NBS at the Lublin City Scale

The application of the excluded criterion showed that within Lublin, five types of NBS were detected and measured on the base of remote-sensed images and GIS approaches. They were (1) urban parks, (2) urban forests, (3) urban water, (4) allotment gardens (equivalent to urban gardens), and (5) wooded areas (equivalent to green strips, green transport tracks, and street trees). They constituted both forms of land use and landscape-scale NBS interventions based on the use of existing elements of GBI. Except that they were created long before the concept of NBS was introduced, they were proven to provide a set of social, environmental, and economic benefits and refer to SDGs (Table 3). Its cost and resource efficiency, however, is quite limited, especially in relation to urban parks, water bodies, and gardens. Given that for an action to be regarded as an NBS, it is not always necessary that 100% of the ingredients are present but merely a majority [7], actions with partial effectiveness were also included in the subsequent part of the research.

### 3.2. Spatial Detection of Identified NBS within the Study Area

According to the adopted approach which ensure the provision of multiple benefits, both to the environment and citizens being the core idea of the NBS concept, a specific set of sizes and accessibility criteria were adopted for each of the five specific forms of NBS identified at the first stage of the research (Table 4). As a result, it was revealed that five analysis types of NBS occupy nearly 20% of the city's area (Figure 3). The largest area and thus the highest percentage share comprised patches of UFs (60% of the NBS area, that is, 12% of the city area) (Table 5). This type, however, was characterized by the smallest number of patches (8), with a significant mean area amounting to 219.54 ha. The smallest area and thus the lowest percentage shared comprised patches of UPs (3.4% of the NBS area, that is, 0.67% of the city area). The share of others classified as type 2 NBS was quite similar and amounted to approximately 2% of the city's area. AGs and Ws featured the highest numbers of patches (70 and 71, respectively), with average sizes of 6 and 3.58 ha, respectively. In addition, patches constituting AGs together with patches of UPs featured similar areas (AGs: AREAmin = 6.00; S.D. = 8.15; UPs: AREAmin = 8.26; S.D. = 6.37).

**Table 4.** Size and accessibility criteria ensuring the provision of multiple benefits to the environment and citizens by analyzing nature-based solutions (NBS) using the existing elements of green and blue infrastructure (GBI).

| **NBS Based on Urban Water Bodies (UW)** |
|---|
| Size criterion: polygons marked as urban water bodies in BDOT classification of the minimum area of 500 m$^2$ (0.05 ha; adopted by the authors due to the lack of formal and methodological foundations);<br>Accessibility criterion: water easily accessible to every citizen (private ponds next to houses were excluded) |
| **NBS Based on Urban Forests (UFs)** |
| Size criterion: polygons of a minimum area of 1000 m$^2$ (0.1 ha; according to the minimum area of forest patch in Polish regulation);<br>Accessibility criterion: areas of public forests, with no access restrictions |
| **NBS Based on Allotment Gardens (AGs)** |
| Size criterion: all plots of allotment gardens according to BDOT classification of LC forms;<br>Accessibility criterion: AGs, according to Polish regulations, are private gardens and thus are available to owners and their families |
| **NBS Based on Urban Parks (UPs)** |
| Size criterion: all polygons of UPs according to the BDOT classification of LC forms with a predominance of grassy or semi-permeable surfaces (min. 70% of the area)<br>Accessibility criterion: all UPs are open to citizens |
| **NBS Based on Wooden Areas (Ws)** |
| Size criterion: polygons of the minimum area of 100 m$^2$ and maximum area of 1000 m$^2$ (0.1 ha)<br>Accessibility criterion: rows of trees along roads, rows of trees along rivers, green squares within or next to residential areas |

**Note:** According Polish regulation forest are area of land dominated by trees of the minimum area of 1000 m$^2$ (0.1 ha). Wooden areas are remaining areas of land dominated by trees of area less than 0.1 ha and higher than 0.01 ha.

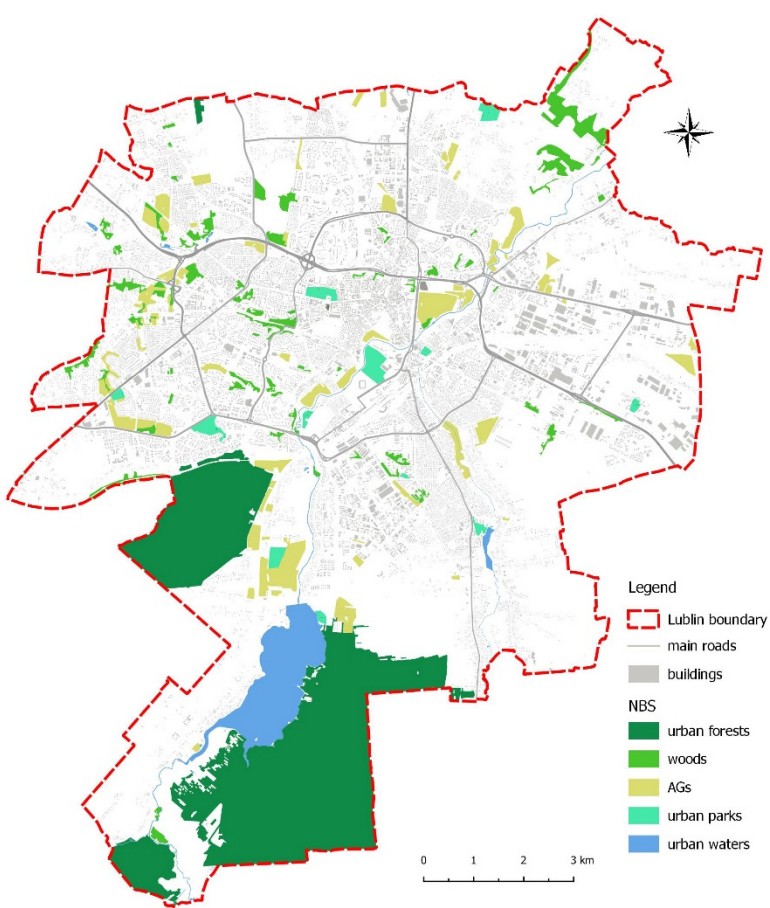

**Figure 3.** Localization and typology of NBS detected within the study area.

**Table 5.** Spatial characteristic of NBS detected at the Lublin city scale.

| | Number of Patches | Total Area (ha) | % of Area in Relation to Are of All the NBS Types | % of Lublin Area | Mean Area (ha) | Area Standard Deviation (S.D.) |
|---|---|---|---|---|---|---|
| Urban parks | 12 | 99.16 | 3.41 | 0.67 | 8.26 | 6.37 |
| Urban forests | 8 | 1756.28 | 60.47 | 11.91 | 219.54 | 385.89 |
| Urban waters | 32 | 347.57 | 11.97 | 2.36 | 10.86 | 59.07 |
| Allotment gardens | 70 | 420.22 | 14.47 | 2.85 | 6.00 | 8.15 |
| Wooden areas | 71 | 280.98 | 9.67 | 1.91 | 3.58 | 6.51 |
| Sum: Area (ha) | | | 2904.21 | | | |
| Sum: % of Lublin area | | | 19.69 | | | |

### 3.3. Calculation of ELQ Indicators

Analysis of landscape-based indictors showed that NBS types contributed differently to the ecological quality at the landscape level. Regarding the maintenance of ecological processes and ecological stability, the biggest UFs had the highest potential. They featured the highest values of indicators, such as PLAND (11.91), MPA (219.54) and LPI (7.91). These values indirectly indicated that the analyzed type of LC possessed a sufficient total habitat area for the existence of species, such as birds and predatory mammal species [30,31]. In addition, patches of UFs were characterized by a 98% contribution of biologically active areas, including both natural (nature reserve) and semi-natural LC forms (Table 6). The lowest contribution to the maintenance of ecological stability features Ws of relatively low MPA (3.58 ha) and the highest ED (7.81), as well as the most disperse spatial structure (COHESION = 95.41). Therefore, from the spatial perspective, except for a few species of birds and synanthropic plants, they do not constitute a potential habitat for a wider group of organisms.

**Table 6.** Results of ELQ indicator calculation.

| Indicator Abbreviation | Urban Parks | Urban Forests | Urban Waters | Allotment Gardens | Wooden Areas |
|---|---|---|---|---|---|
| MPA [ha] | 8.26 | 219.54 | 10.86 | 6.00 | 3.58 |
| PLAND [%] | 0.67 | 11.91 | 2.36 | 2.85 | 1.91 |
| BAA [%] | 88.12 | 98 | 100 | ~90 * | 95.15 |
| $AT_{natural}$ [%] | 0.00 | 1.43 | 0.00 | 0.00 | 0.00 |
| $AT_{semi\text{-}natural}$ [%] | 0.00 | 96.57 | 10.93 | 0.00 | 0.00 |
| $AT_{antro\ no\ 1}$ [%] | 88.12 | 0.00 | 89.07 | 90.00 | 95.15 |
| $AT_{antro\ no\ 2}$ [%] | 11.88 | 2.00 | 0.00 | 10.00 | 4.85 |
| ECOLBAR [m/ha] | 0.00 | 6.87 | not applicable | 5.97 | 3.21 |
| PD [nos/100 ha] | 0.08 | 0.07 | 0.20 | 0.47 | 0.47 |
| ED [m/ha] | 1.26 | 3.72 | 6.61 | 6.76 | 7.81 |
| LPI [%] | 0.16 | 7.91 | 2.30 | 0.38 | 0.38 |
| FRAC_MN [-] | 1.07 | 1.11 | 1.13 | 1.09 | 1.15 |
| COHESION [-] | 96.87 | 99.69 | 99.89 | 96.75 | 95.41 |
| SLOPE [%] | Max 20 Average 3 | Max 36 Average 2 | Lack of data | Max 5 Average 1 | Max 48 Average 8 |

* Calculated based on data from Sowińska-Świerkosz et al. [12] based on the mean area of AGs and mean area of cabin and paved surfaced.

In the diversity analysis at the landscape level, AGs and Ws had the highest contribution, as they were characterized by many patches (70 and 71, respectively), the highest density (PD = 0.47), and dispersed locations (COHESION = 96.75 and 95.41, respectively) within the entire city structure. Geomorphological indicators, including SLOPE, have shown the high contribution of Ws to diversity; varied topography promoted the existence of different plant species [51]. Due to the aggregated character of UF patches, they had a low impact on landscape-level diversity.

The analysis of continuity of ecological structures showed the negative impact of ecological barriers in the case of UPs and Ws (ECOLBAR = 0.00 and 3.21, respectively).

In addition, Ws and AGs were evenly distributed within the study area, thus positively contributing to the continuity of greenery. The positive impact of the latter, however, was affected by the high density of paved roads crossing garden structures (ECOLBAR = 5.97). The same applied to UFs with the highest density of ecological barriers (ECOLBAR = 6.87).

Regarding the aesthetic landscape values, similar values of FRAC_MN approaching '1' indicated that the shape of the analyzed NBS types was approaching square, which is characteristic of city structures dominated by rectilinear man-made elements. However, BAA and AT indicators showed that due to the high share of natural and semi-natural green and blue areas, which are generally positively perceived by people, UFs were the highest contributors to aesthetic value; conversely, Ws were the lowest.

## 4. Discussion

### 4.1. Elements of Urban GBI as NBS

Within Lublin, five types of GBI were identified as type 3 NBS—design and management of artificial or semi-natural ecosystems. These elements fulfill most of the formal requirements to recognize an intervention as NBS proposed by Sowińska-Świerkosz and García [7] (Table 3). Previous research conducted by other authors in relation to different elements of GBI, including UPs [21,37,38], UFs [40–42], UWs [42–45,52], AGs [16,46–48], and Ws [11,49,50], showed that these elements provide and/or improve environmental, social, and economic benefits, such as recreational and spiritual or cultural ecosystem services, physical and mental health, carbon sequestration, air purification, noise reduction, biodiversity and, in the case of AGs and UWs, food provisioning. In addition, the analyzed elements of GBI were found to mitigate various global problems, reflecting the SDGs, primarily good health and well-being, sustainable cities, responsible consumption, climate actions, and life on land and in water [53]. A more complex issue, however, was the aspect of GBI's cost and resource-efficiency. To be regarded as an NBS, an action should be both

cost-efficient (meaning that it produces good results without costing a lot of money) and resource-efficient (meaning that it uses building materials, natural resources, and energy in a sustainable manner, while having minimal impact on the environment) [5,54]. Furthermore, the cost of a solution's implementation, maintenance, or transformation should not exceed the potential benefits [1,3]. An understanding of the economic efficiency of most of the analyzed elements of GBI in Polish conditions remains limited. For example, UGs located in Lublin require continuous planting, watering, and mowing; moreover, most Polish AGs were not equipped with renewable sources of energy [12,49]. Therefore, the analyzed elements of GBI were considered as NBS, of which cost efficiency was limited. To unambiguously define whether a given element should be classified as a NBS, each element should be analyzed separately based on detailed criteria, for example, by calculating the NBS effectiveness indicators proposed by Sowińska-Świerkosz and García [3], which include the level of social acceptance, perceived level of aesthetic value, existence of political support and guidance, existence of rainwater recovery devices, amount of energy produced from renewable sources, and carbon and heat absorption capacity. Such an approach is necessary, as possessing all major features linked to NBS does not axiomatically render elements of GBI a successful ecosystem-based solution. We agreed with Nesshöver et al.'s [8] conclusion that, despite the fact that GBI and NBS are based on the use of nature and are directed to provide various benefits, the biggest difference between these concepts pertains to the difference between the terms "infrastructure" and "solution." GBI refers to the green and blue structures needed for a society or enterprise to operate, and NBS, as defined by the EC, should solve the encountered problem(s) [7]. Therefore, challenges should be detected a priori and constitute the main reason for NBS implementation or reshaping and modernizing existing green infrastructure [8]. As a result, GBI can be considered a NBS if it contributes to solving the encountered problem(s) [11] and has a high level of economic efficiency [7]. Therefore, to be congruent with the concept of NBS, the analyzed elements of (historical) GBI should be somehow altered, for example, by enlarging their area, utilizing devices dedicated to the use of rainwater, linking renewable sources of energy, and implementing NBS projects [12].

### 4.2. Impact of Analysed NBS Types on Ecological Landscape Quality

The analysis clearly revealed that in the Lublin city scale, different types of NBS contribute in varied ways to improving ecological quality at the landscape level. Among them, UFs and Ws proved to be of the greatest importance (Figure 4). UFs, due to their compactness, were found to have the most positive impact on the maintenance of ecological processes and ecosystem stability; however, the same spatial characteristic indirectly explained their low impact on diversity and continuity of ecological structure at the landscape scale. UFs also proved to be particularly important aesthetically, as natural LC forms are considered to have the highest perceived value, followed by semi-natural forms [55]. Additionally, the full assessment of the aesthetic level of landscape quality is mainly subjective, and objective measures applied in this study constitute only approximate values. Ws also proved to have a different impact depending on the analysis level of EQ. These areas are crucial for the maintenance of a high level of diversity and the continuity of ecological structure at the landscape scale, as seen within the entire city spatial structure. The low mean area of patches, however, indicates that they are not capable of maintaining important ecological processes and have a significant positive impact on ecosystem stability. AGs, which were found to be evenly distributed across the study area, positively contributed to the continuity of ecological structure and diversity on the landscape scale [12]. The positive impact of AGs, however, was found to be affected by the rather high density of paved roads and the homogeneous relief of the land. Surprisingly, the results indicated that UWs had no outstanding impact on ELQ. This is primarily due to the high level of anthropogenic transformation of the two main water bodies within the Lublin city structure: the Zemborzycki Reservoir (which is a dam reservoir) and the Bystrzyca River valley (which is subjected to strong anthropopressure) [56]. UPs also showed no specific impact

on the analyzed dimensions of ELQ, due to the small number of parks and their modest total area, as well as the relatively low share of biologically active areas. Therefore, the conclusions are limited to this case study, and in other areas, the impact of UWs may be considerably higher.

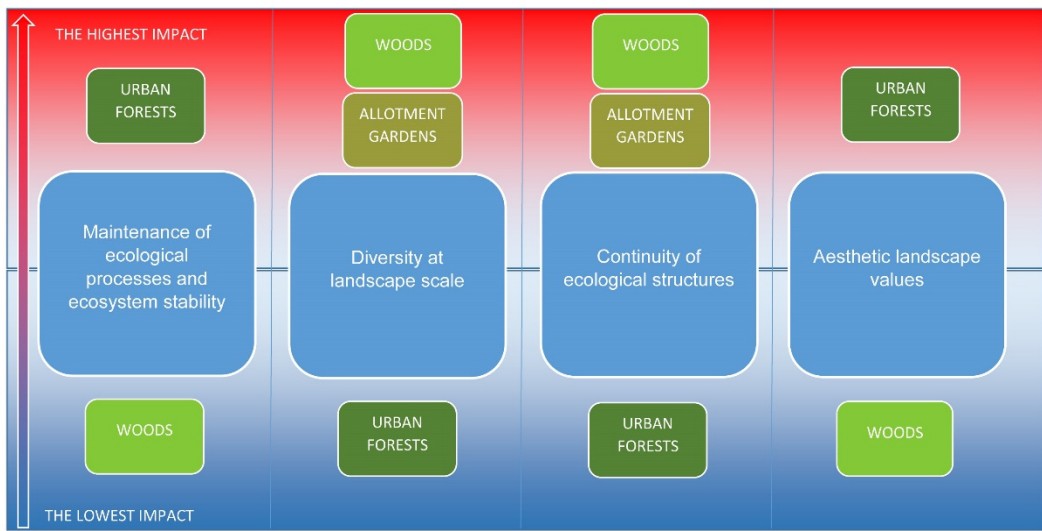

**Figure 4.** Ideogram of analyzed types of NBS based on the GBI impact on Ecological Landscape Quality.

However, the conclusions were formulated based only on landscape-based surrogates, which indirectly explains the ecological state of a given area based on the application of substitute data. They provide only approximate information on ecological value, although they are of great importance when other data types are not available or are solely available at a high cost and with considerable time and effort [31,51]. An overall conclusion on the impact of different types of NBS on ELQ must be based on the application of a set of data sources and methods, including in-situ assessment of soil, water, air, and plant quality. Furthermore, the aesthetic level of landscape quality is mainly subjective, and the objective measures applied in this study constitute only approximate values. The usefulness of different kinds of landscape-based surrogates, however, has been demonstrated in previous studies assessing the effectiveness of NBS projects [9,26,57,58]. As there is a lack of available data to calculate a total set of indicators crucial for a comprehensive assessment of NBS' environmental impacts [1,8,57], the result of this study may be treated as a rough estimate of the actual impact.

## 5. Conclusions

The analysis clearly revealed that on the Lublin city scale, different NBS types based on the elements of GBI contribute in varied ways to improving ecological quality at the landscape level. Among them, however, UFs and Ws were of the greatest importance. UWs had no outstanding impact on ELQ, primarily because of the high level of anthropogenic transformation in the two main water bodies within the Lublin city structure. Therefore, the conclusions were limited to this case study, and in other areas, the impact of UWs may be considerably higher.

The results were formulated based only on landscape-based surrogates, which provide only approximate information on ecological values. An overall conclusion on the impact of different NBS types on ELQ must be based on the application of a set of data sources and methods, including in-situ assessment of soil, water, air, and plant quality.

In relation to the development of knowledge on the difference between the concepts of GBI and NBS, it should be concluded that it pertains to the difference between the terms "infrastructure" and "solution." GBI can be considered a NBS if it contributes to solving the encountered problem(s) and has a high level of economic efficiency; not all green and

blue urban structures should automatically be considered an NBS. Therefore, the size, accessibility, and benefits provision criteria adopted in the paper to recognize whether a given element of GBI can be considered a type 3 NBS should be extended by the fourth criterion—its efficiency in solving a given environmental problem or tackling a given social challenge.

**Author Contributions:** Conceptualization, B.S.-Ś. and J.W.-M.; methodology, B.S.-Ś.; software, J.W.-M. and M.M.-Ś.; validation, B.S.-Ś., J.W.-M. and M.M.-Ś.; formal analysis, B.S.-Ś.; investigation, B.S.-Ś.; resources, B.S.-Ś., J.W.-M. and M.M.-Ś.; data curation, B.S.-Ś., J.W.-M. and M.M.-Ś.; writing—original draft preparation, B.S.-Ś.; writing—review and editing, M.M.-Ś.; visualization, M.M.-Ś.; supervision, B.S.-Ś.; project administration, M.M.-Ś.; funding acquisition, B.S.-Ś. All authors have read and agreed to the published version of the manuscript.

**Funding:** This research received no external funding.

**Institutional Review Board Statement:** Not applicable.

**Informed Consent Statement:** Not applicable.

**Conflicts of Interest:** The authors declare no conflict of interest.

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
