# Peer review of "An Assessment of the Ecological Landscape Quality (ELQ) of Nature-Based Solutions (NBS) Based on Existing Elements of Green and Blue Infrastructure (GBI)"

_sustainability, doi:10.3390/su132111674_

Round 1

Reviewer 1 Report

Sustainability 1365786

A well-written article but needs work to be publishable.

Introduction

Lines 57-61 authors need a better definition of ecological landscape quality and why it is important to assess.

Lines 84-35 Study objectives –need to define NBS actions.

Materials and Methods

Line 91 – please explain - why not a detailed assessment of each landscape element.

Lines 92-95 there are a lot of assumed linkages or assumptions in this section. How is it supported?

Table 2: line 126+

It is unclear to this reviewer what the difference is between Urban Forests and Wooden areas. Are not Wooden Areas Urban Forests- why the distinction?

Lines 127-128 It is unclear how these indicators are derived other than” non-systematic review of previous papers”. It would be better to have some discussion regarding the linkage between the indicators and the four main aspects for assessment of ecological quality.

Table 3 It would be better to have some discussion of the rationale for choosing and using these indicators.

Results

Lines 145-149 the meaning is unclear- please explain.

Lines 166 on under Calculation of ELQ indicators- It would be better to spell out cover types throughout this section

Discussion

Line 208 –please explain “unsophisticated types of NBS”.

Lines 214-281 Are very important and should be in a conclusion section.

Missing conclusion

The paper needs a conclusion, which as a minimum should include:

What did we learn and how generalizable is it?

What are the study limitations?

What future research is needed?

Reviewer 2 Report

Dear all,

Your work deals with a very relevant and actual issue.

In fact, it was pretty interesting to read this working manuscript!

Even if this work presents a considerable quality in the actual status, some improvements should be considered:

  • a conclusion section should be added
  • Also, a sub-section regarding this study's limitations and future research lines should be added.

Best,

Reviewer 3 Report

The article entitled "An Assessment of the Ecological Landscape Quality (ELQ) of Nature-Based Solutions (NBS) Based on Existing Elements of Green and Blue Infrastructure (GBI)" deals with an important topic of the current research that is the biophysical measure of the benefits provided by NBS thought the green infrastructures in cities. Although this argument is well debated and the methodologies for this identification refers more to the ecosystem service modelling by spatial assessment, the authors decided to perform a traditional landscape composite index based on a list of indicators. To help the authors in re-structuring this work, I divide the main problems into two areas: theoretical and (as consequence) empirical.

On the theoretical side, the introduction of this work is missing crucial elements, such as a proper definition of GI, the identification of NBS (which are not land cover types as assumed by this manuscript), the fact that GI are based on multifunctional ecosystem services which are not detected by this paper, and the partial association with SDG and the costs of implementation which are not developed at all in the text.

Then, within these premises, the empirical argumentation lacks of clarity: 1. this paper refer to previous work which has not been explained; 2) the method is hidden since there is no explanation of the GIS processing and the kind of material used for producing the landscape indicators; 3) these indicators are not useful in their comparison between different land-use classes... slope or fragmentation or patch size are important indicators of the quality of the landscape if analyzed for the same land use... or you can use all these indicators to characterize the different land uses... but they don't tell you anything about the ecosystem provision. An urban park of 0.5 ha can store more carbon in the soil of an outdoor grassland... and the only way to measure this parameter it to map the biophysical status of ES; 4) your discussion are mainly based on ecosystem assumptions that are not demonstrated; 5) conclusions are missing.

I suggest you re-formulate this work while strongly focusing your discourse around the Characters of the green areas in your case of study. Within this approach, you can save your method and make a coherent and logical work.

You can find my detailed comments in the attached file.

Good luck!

Round 2

Reviewer 1 Report

The changes made to explaining methodology and context greatly improve the manuscript- however there are numerous spelling and English usage grammar issues throughout the manuscript that need attention.

Author Response

The changes made to explaining methodology and context greatly improve the manuscript- however there are numerous spelling and English usage grammar issues throughout the manuscript that need attention.

The paper has been send again to proof-reading service.

Reviewer 3 Report

I see that author made moderate changes here and there.

Nevertheless, the main theoretical mistake of this work remained untouched... what you call NBS are not NBS. Please try to deal with this issue, otherwise, this work cannot be scientifically acceptable.

Please explain in the method what kind of geoprocessing method did you followed for your landscape quality variables... your description is too general and vague.

Pay attention to spacing in the text.

Attached are my detailed comments.

Good luck.

Author Response

Thank you for all your comments. We appreciate the time and effort put into the review of our paper. We are sure that the manuscript has been significantly improved thanks to your suggestions.  Some of your comments, however, were not included into the revision. Below, we provide detailed explanation.

Round 3

Reviewer 3 Report

I see that the authors worked on the suggested direction.

Just take a look and correct the few final comments (on the file attached).

Good luck!

Author Response

Many thanks to the reviewer for his/her thorough review and insightful comments on this paper. All comments were taken into account into text. The final version of the manuscript is definitely more valuable.